# Degradation of a New Herbicide Florpyrauxifen-Benzyl in Water: Kinetics, Various Influencing Factors and Its Reaction Mechanisms

**DOI:** 10.3390/ijms241310521

**Published:** 2023-06-23

**Authors:** Rendan Zhou, Zemin Dong, Long Wang, Wenwen Zhou, Weina Zhao, Tianqi Wu, Hailong Chang, Wei Lin, Baotong Li

**Affiliations:** 1College of Land Resources and Environment, Jiangxi Agricultural University, Nanchang 330045, China; rendanzhou@stu.jxau.edu.cn (R.Z.); zemindong2020@stu.jxau.edu.cn (Z.D.); wl2283807483@stu.jxau.edu.cn (L.W.); w475172773@stu.jxau.edu.cn (T.W.); hailongchang@stu.jxau.edu.cn (H.C.); linwei0035@stu.jxau.edu.cn (W.L.); 2Jiangxi Agricultural Technology Extension Center, Nanchang 330046, China; 13767120850@163.com; 3College of Food Sciences, Jiangxi Agricultural University, Nanchang 330045, China; wenwenzhou@jxau.edu.cn

**Keywords:** disposable face masks (DFMs), environmental behavior, florpyrauxifen-benzyl, hydrolysis, microplastics (MPs), pathway

## Abstract

Florpyrauxifen-benzyl is a novel herbicide used to control weeds in paddy fields. To clarify and evaluate its hydrolytic behavior and safety in water environments, its hydrolytic characteristics were investigated under varying temperatures, pH values, initial mass concentrations and water types, as well as the effects of 40 environmental factors such as microplastics (MPs) and disposable face masks (DFMs). Meanwhile, hydrolytic products were identified by UPLC-QTOF-MS/MS, and its hydrolytic pathways were proposed. The effects of MPs and DFMs on hydrolytic products and pathways were also investigated. The results showed that hydrolysis of florpyrauxifen-benzyl was a spontaneous process driven by endothermic, base catalysis and activation entropy increase and conformed to the first-order kinetics. The temperature had an obvious effect on hydrolysis rate under alkaline condition, the hydrolysis reaction conformed to Arrhenius formula, and activation enthalpy, activation entropy, and Gibbs free energy were negatively correlated with temperature. Most of environmental factors promoted hydrolysis of florpyrauxifen-benzyl, especially the cetyltrimethyl ammonium bromide (CTAB). The hydrolysis mechanism was ester hydrolysis reaction with a main product of florpyrauxifen. The MPs and DFMs did not affect the hydrolytic mechanisms but the hydrolysis rate. The results are crucial for illustrating and assessing the environmental fate and risks of florpyrauxifen-benzyl.

## 1. Introduction

Pesticides are the mainstay to prevent and control crop diseases and pests in agricultural production, and play an important role in ensuring the quality of agricultural products, food safety, and public health safety [1,2]. However, most pesticides are released into the environment no matter how carefully they are applied, except for a small amount for crops. Then they will be migrated, transformed, and accumulated in the environment, resulting in a certain harm to the structure and function of the ecosystem and finally posing a serious threat to human health through food chain. Among them, residual pesticides usually enter the water environment through rainwater flushing, surface runoff, or leaching with soil pore water [3,4,5].

Numerous studies have confirmed that a large variety of pesticide residues with high concentrations were detected in surface waters worldwide. Xu et al. [6] investigated pesticide pollution in surface waters of major river basins in China, and nine pesticides were detected in 27 sampling points, among which the detection rate of atrazine was 100%. Peng et al. [7] identified a maximum concentration of 1726 ng L^−1^ of atrazine in surface water from the Yangtze River Delta in China. Stone et al. [8] reported that 11 herbicides, 4 insecticides, and 1 fungicide were detected in 39 important rivers in United States. Souza et al. [9] reviewed pesticides monitoring studies of surface waters worldwide, showing that atrazine and its metabolites, metolachlor, chlorpyrifos, and tebuconazole were largely present, followed by diuron, dimethoate, and carbendazim with high concentrations and frequencies. They found that developing countries existed a wide variety and higher concentrations of pesticides in surface waters as compared with developed countries. The intake of water contaminated with pesticides may cause multifarious health problems, for example, cancer, cardiovascular diseases, neurological diseases, respiratory ailments, reproductive problems, and diarrhea [3,10]. The pollution of pesticides to the environment has attracted more and more attention, and their environmental behaviors and ecotoxicity have increasingly became the focus of environmental science research.

The environmental behaviors of pesticides including soil degradation, hydrolysis, photolysis, sorption and desorption, and leaching will affect the stability and effectiveness, as well as their residual transfer and safety evaluation in the environment [11]. Among them, hydrolysis was one of the main dissipation approaches for much pesticide degradation in the natural environment. The essence of pesticides hydrolysis is nucleophilic substitution reaction, i.e., the nucleophilic groups (H_2_O or OH^−^) in aqueous solution attack central electrophilic groups (C, N, S, P, etc.) of the pesticides molecules and replace the leaving groups, and generating various hydrolytic products. Meanwhile, the hydrolytic process is affected by many factors, including the characteristics of pesticides and environmental factors (such as pH, water temperature, and clay minerals) [12]. The hydrolytic behavior of pesticides and influencing factors were beneficial to evaluate the residual characteristics in water and the impact on aquatic ecological environment. Moreover, hydrolysis reactions did not always thoroughly generate less toxic substances, but sometimes generate highly toxic intermediates, which posed a significant potential threat to aquatic organisms. For example, the hydrolytic product of methyl parathion was *p*-nitrophenol with highly toxic and carcinogenic, which was difficult to biodegrade and a priority-controlled pollutant for the United States Environmental Protection Agency [13]. The dichlorvos was a decomposition product of trichlorfon, which was a typical genetic mutagen and listed as carcinogen, possessing over 10 times the toxicity of trichlorfon [14]. In addition, the molecular mass of degradation products is usually smaller than that of the parent compounds, which may aggrandize their migration potential, posing a significant potential threat to non-target organisms [12]. Therefore, the analysis and identification of hydrolytic products of pesticides are of great significance to the evaluation of potential risks, and the rational and safe application in the agricultural production.

With the development of agricultural technology, plastic film mulching has been widely used as a superior measure to increase agricultural yield. However, due to the poor management, a large amount of extensive, persistent, and toxic microplastics (MPs) contaminants are widely distributed in farmland environment worldwide [15]. The MPs have the characteristics of small volume, large specific surface area, and strong hydrophobicity, etc., which can also be used as carriers of toxic metals, microorganisms, antibiotics, and pesticides [16]. The MPs have also been detected in water environments worldwide [17,18] and even found in human blood [19], and their ubiquity has become a key environmental issue of global concern. Meanwhile, the global use of disposable face masks (DFMs) increased exponentially with the outbreak and widespread of the COVID-19 pandemic. DFMs are plastic products that do not biodegrade easily. They have entered the natural environment, freshwater system, and ocean because they were not effectively recycled, posing enormous threat to the environment [20].

Florpyrauxifen-benzyl is a novel pyridine-2-carboxylate auxin rice herbicide mainly developed by Dow AgroSciences, which belongs to a new category of synthetic hormone herbicide. It can interfere with the normal physiological and biochemical functions of weeds by binding with hormone receptors and achieving the effect of weeding. It has a special effect for controlling *Echinochloa crusgalli*, which is difficult to prevent in paddy fields [21]. The current research has mainly focused on its synthesis and development, residual analysis, and field control effect, but the migration, transformation, and toxicology in the environment have seldom been reported [21,22,23,24,25,26]. In our previous work, its residue analysis method and dissipation behavior in a natural paddy field environment were investigated with dissipation half-lives of less than 3 d, which did not cause persistent residues [27]. Because a series of environmental behaviors of pesticides may affect the residue in the environment, it is extremely important to investigate their environmental behaviors to better evaluate the persistence and safety [28].

Considering the hydrolytic behavior of florpyrauxifen-benzyl has seldom been reported, the hydrolytic characteristics were investigated under varying temperatures, pH values, initial mass concentrations, and water types by static simulation experiments to clarify hydrolytic kinetics. The effects of environmental factors such as MPs, DFMs, fertilizers, cations, anions, surfactants, coexisting herbicide, humic acid, and biochar on hydrolysis were also investigated. Meanwhile, the possible hydrolytic products were analyzed and identified by UPLC-QTOF-MS/MS to speculate the hydrolysis pathways, as well as the effects of MPs and DFMs on hydrolytic products and pathways. These results will provide reference for accurate and comprehensive assessment of florpyrauxifen-benzyl residual behaviors in water environments, and enable a more sustainable and safer application in agro-ecosystem.

## 2. Results and Discussion

### 2.1. Hydrolytic Characteristics of Florpyrauxifen-Benzyl

The degradation behavior of pesticides in water is one of the main aspects of their non-biological degradation. The comprehensive and intensive understanding of the pollution regulation and hydrolysis ability are important to evaluate the safety of pesticides on ecological environment. The hydrolysis of florpyrauxifen-benzyl was investigated at varying temperatures, pH values, initial mass concentrations and water types, and hydrolysis kinetics parameters were shown in Table 1. It was found that the hydrolytic dynamics conformed to the first-order kinetics, and the coefficient of determination (*R*^2^) was 0.8538–0.9875. Both the hydrolysis rate constant (*k*) and half-life (*T*_0.5_) can be used to evaluate the hydrolysis rate, which were 0.0031–4.6981 d^−1^ and 0.15–220.75 d, respectively. The high temperature significantly increased hydrolysis rate of florpyrauxifen-benzyl under neutral and alkaline conditions. The hydrolysis was faster in the alkaline condition than the neutral condition at the same temperature, and more easily affected by temperature in alkaline condition. However, hydrolysis was very slow under acidic conditions, the *T*_0.5_ was more than 120 d, and hydrolysis rate was almost unaffected by temperature. The results showed that the hydrolysis rate of florpyrauxifen-benzyl in the descending order was as follows: alkaline solution, neutral solution and acid solution. The high temperature and alkaline condition were favorable for hydrolysis of florpyrauxifen-benzyl, while low temperature and acidic condition were unfavorable, indicating that the reaction was more effectively catalyzed by hydroxide ions than hydronium ions or neutral water molecules.

In pH = 9 alkaline solutions, the hydrolysis dynamics of florpyrauxifen-benzyl at different temperatures were shown in Figure 1A. The temperature effect coefficient (*Q*), activation energy (*E*_a_), activation enthalpy (Δ*H*), activation entropy (Δ*S*) and Gibbs free energy (Δ*G*) were shown in Appendix A. The hydrolysis rate of florpyrauxifen-benzyl rapidly increased with the increasing temperature (15–50 °C), and it could be completely hydrolyzed at 50 °C with a *T*_0.5_ of only 0.15 d. It may be the hydrolysis reaction of pesticides was an endothermic reaction and its *E*_a_ mainly generated from the collisions between molecules [29]. The increasing temperature would accelerate dramatically the molecular movement, thus increased the depolymerization and chain-breaking of pesticides molecules [12]. Plotting ln*k* vs *T*^−1^ gave a straight line with good linear relationship (*R*^2^ = 0.9995), as shown in Figure 1B, indicating that the reaction conformed to Arrhenius formula. The *E*_a_ calculated from the slope was 140.0197 kJ mol^−1^. The *E*_a_ was one of the determining factors of the reaction speed and affected hydrolysis rate of florpyrauxifen-benzyl. The higher the *E*_a_, the higher the energy required for the molecules to collide, so the slower the reaction rate. Meanwhile, the *E*_a_ also reflected effect of temperature on *k*, and the *k* increased significantly with the increasing temperature under a higher the *E*_a_; otherwise, the change was not obvious. The *k* of florpyrauxifen-benzyl was greatly affected by temperature [30].

The *Q* is usually used to illustrate the relationship between *k* and temperature. The average *Q* of florpyrauxifen-benzyl was 6.4634, indicating that temperature had a significant effect on its hydrolysis rate under alkaline condition. Compared with rule of van’t Hoff, namely, the reaction rate increases 2–4 times for every 10 °C increase in temperature. The *Q* of florpyrauxifen-benzyl was higher, which might be due to the higher *E*_a_ of the reaction [31]. In the process of chemical reaction activation, if the activation complex molecules had more freedom degrees than the reactant molecules, the ring structure of the compound would be destroyed, the degree of solvation would be reduced, the rigid structure would become relaxed structure, and Δ*S* would be greater than zero [31,32]. The Δ*S* was 0.2950–0.2960 kJ (mol K)^−1^ with the low degree of system disorder, indicating that hydrolysis reaction of florpyrauxifen-benzyl was mainly driven by the increasing activation entropy [31]. The relationships between Δ*H*, Δ*S*, Δ*G* and temperature were shown in Figure 1C–E, which were decreased with increasing temperature, showing a significant negative correlation. 

The hydrolysis rates of florpyrauxifen-benzyl with different initial mass concentrations in aqueous solution were 1 mg L^−1^ > 2 mg L^−1^ > 5 mg L^−1^. The *k* (0.0409–0.0039 d^−1^) decreased but *T*_0.5_ (16.96–176.82 d) increased gradually with the increasing initial concentrations. It may be that florpyrauxifen-benzyl molecules had more chances to contact with OH^−^ in aqueous solution when florpyrauxifen-benzyl concentration was low, so the hydrolysis rate was higher; while florpyrauxifen-benzyl molecules had less chance to contact with OH^−^ in aqueous solution with the increasing concentration, resulting in decrease in the hydrolysis rate [33]. The hydrolysis rates of florpyrauxifen-benzyl in different water types were ultrapure water > lake water > seawater > paddy water > tap water, and the *T*_0.5_ were 16.96, 24.13, 25.66, 29.11 and 48.04 d, respectively. The *k* in ultrapure water was about 1.42, 1.51, 1.72, 2.83 times that of lake water, seawater, paddy water and tap water respectively. Because 5 kinds of water all had been sterilized, the influence of microorganisms on hydrolysis was excluded. The possible reason was that natural water contains a large number of dissolved substances, microelements and suspended particulate matters which may change the pH through sorption catalysis or the formation of complex to impact hydrolysis of florpyrauxifen-benzyl [31]. Moreover, some dissolved organic matter in natural water could inhibit hydrolysis of some pesticides. This was because those soluble organic matters mainly existed in the form of anions, which hindered the formation of anion transition state in the hydrolysis process of pesticides [31]. Therefore, the hydrolysis of florpyrauxifen-benzyl in natural water is more complex, which caused by interaction of microorganisms and various soluble substances in water. 

### 2.2. Effect of Environmental Factors on Hydrolysis of Florpyrauxifen-Benzyl

There are many factors influencing hydrolysis of pesticides. In addition to the physical and chemical properties of pesticides and the temperature and pH value of solution, the presence of microbial activity, inorganic and organic matters, metal ions, clay minerals and other substances in different water environments could affect hydrolysis process through sorption, catalysis, enrichment, formation of complexes and other functions [12,34].

#### 2.2.1. MPs

The effects of 12 kinds of common MPs with different contents on the hydrolysis were shown in Table 2 and Figure 2. The *k* and *T*_0.5_ of blank control group were 0.0480 d^−1^ and 14.43 d, respectively. The hydrolysis of florpyrauxifen-benzyl was affected by the addition of a certain amount of different kinds of MPs. The *k* and *T*_0.5_ were 0.0320–0.2334 d^−1^ and 2.97–21.69 d, respectively. The PA, PS, LDPE, PP and PMMA promoted hydrolysis, but the promoting ratio of LDPE was not obvious. The PHB, PBS, PBAT, PHA, PLA, PVC and PE inhibited hydrolysis at low contents while promoted hydrolysis at high contents. The promoting ratio on hydrolysis was as high as 385.99% with 0.50% of PBAT while the inhibiting ratio was −33.47% with 0.050% of PHA. Overall, promoting effect was more obvious. The possible reason is that a solid-liquid hydrolysis interface was formed between the added MPs and buffer solutions, and the MPs all have the characteristics of large specific surface area and strong hydrophobicity, which can provide enough sorption sites for florpyrauxifen-benzyl molecules, so as to effectively improve the concentration of florpyrauxifen-benzyl molecules around the solid-liquid interface. On the other hand, the MPs contain a large number of functional groups which can adsorb OH^−^ and other nucleophiles by the electrostatic attraction and other forces, and the concentration of nucleophiles around the liquid-solid interface will be increased, thus accelerating the rate of the nucleophilic substitution reaction of florpyrauxifen-benzyl. Hence, hydrolysis was affected by the sorption and enrichment of MPs on florpyrauxifen-benzyl [35,36,37]. In addition, the PBAT contains a large number of hydroxyl, ester, carboxyl and other oxygen-containing groups, which can be used as nucleophiles in the hydrolysis of florpyrauxifen-benzyl, thus significantly improving the hydrolysis rate.

#### 2.2.2. DFMs

The DFMs are multilayered consisting of outer layer, middle layer, inner layer, ear band and pliable noseclip, etc. (Appendix A). The outer layer and inner layer of DFMs used in this test were made from spunbonded non-woven fabric of polypropylene (PP), the middle layer, the ear band and the pliable noseclip were made from melt-blown non-woven fabric of polypropylene (PP), polyester (PET), polyurethane (PU), and polyethylene (PE) wrapped in wire, respectively. The influences of DFMs and its different parts with different contents on hydrolysis were investigated. From Table 2 and Figure 3, the *k* and *T*_0.5_ were 0.0365–0.1168 d^−1^ and 5.93–18.97 d, respectively. The middle layer promoted hydrolysis with 8.43–29.47% of promoting ratios. The outer layer inhibited hydrolysis with inhibiting ratios ranged from −11.79% to −23.91%, and the lower the contents, the more significant inhibiting effect. The whole mask, inner layer and ear band inhibited hydrolysis at low content while promoted hydrolysis at high content, in which promoting effect of hydrolysis by 0.50% of ear band was up to 143.21%. It may be that a solid-liquid hydrolysis interface was also formed between DFMs and its parts and buffer solutions, which was similar to MPs systems. The main components of the DFMs, such as PP, PET and PU, have large specific surface areas and polymer properties, which can provide enough sorption sites for florpyrauxifen-benzyl molecules, so as to effectively improve the concentration of florpyrauxifen-benzyl molecules around the solid-liquid interface. Meanwhile, the DFMs and its parts can also gather nucleophilic reagents around the solid-liquid interface by electrostatic attraction and other forces, thus accelerating the rate of nucleophilic substitution reaction of florpyrauxifen-benzyl [35,36,37]. Furthermore, 0.50% of ear band was the most effective in promoting hydrolysis. There are a large number of ester groups in the molecular chain of PET, and the molecular structure of PU contains -NHCOO- units. These groups might act as nucleophiles in the reaction, thus accelerating the hydrolysis of florpyrauxifen-benzyl. However the outer layer inhibited the hydrolysis, which may be related to its structure and so on, and needs to be confirmed by further research.

#### 2.2.3. Fertilizers

China is a big agricultural country and the use of nitrogen, phosphorus and potassium fertilizers in agricultural production is quite considerable, leading to serious excessive of nitrogen, phosphorus and potassium in water environment in many areas. Therefore, attention should be paid to the pollution of water environment, chemical degradation and efficacy of pesticides caused by chemical fertilizers [38]. As can be seen from Table 2 and Figure 4 that the *k* and *T*_0.5_ of blank control group were 0.0483 d^−1^ and 14.34 d, respectively. The organic fertilizer accelerated the hydrolysis rate of florpyrauxifen-benzyl, and the higher the content of organic fertilizer, the more obvious of the promoting effect. The promoting ratios were 14.16–60.06%, the *T*_0.5_ were shortened from 12.57 to 8.96 d. The urea with low content had a slight inhibiting effect on hydrolysis, and the inhibiting ratios ranged from −0.08% to −6.25%, but 0.50% of urea promoted hydrolysis with a promoting ratio of 10.64%. The CMPF, potash fertilizer, compound fertilizer and OICF inhibited the hydrolysis with inhibiting ratios from −2.52 to −47.50% and *T*_0.5_ of 14.72–27.32 d. The inhibiting orders from strong to weak were approximately compound fertilizer > OICF > CMPF > potash fertilizer. The organic fertilizer promoted hydrolysis, while OICF inhibited hydrolysis. It may be organic fertilizer contains a large number of organic substances, and their surfaces contain abundant groups (such as hydroxyl, carbonyl and carboxyl groups, etc.) that could interact with florpyrauxifen-benzyl, as well as the sorption for florpyrauxifen-benzyl molecules, and thus promoting the occurrence of hydrolysis. The urea is a diamide compound of carbonic acid containing two amino groups, which can make aqueous solution weakly alkaline. It may be that urea with low content did not affect the pH value of solution, while the high content significantly increased pH value, so urea with high content promoted the hydrolysis obviously. The potash fertilizer did not change the pH value of aqueous solution and its inhibiting effect on hydrolysis was the weakest.

#### 2.2.4. Cations

In natural water environment, the pesticides hydrolysis is less likely to be significantly affected by metal ions, but in some special environments where the concentrations of metal ions are high, such effects cannot be ignored. Meng et al. investigated the effects of 0.01–1 g L^−1^ of Cu^2+^, Ni^2+^, Zn^2+^, Pb^2+^ and Fe^3+^ on the hydrolysis of vanisulfane, and found that the hydrolysis rate increased with the increase in concentration of Cu^2+^, while Ni^2+^, Zn^2+^, Pb^2+^ and Fe^3+^ had no significant effect on the hydrolysis rate [39]. Song et al. studied the hydrolysis kinetics of bendazone and found that 1–10 mg L^−1^ of Fe^3+^ could accelerate the hydrolysis rate [40]. The effects of 9 common cations in natural water on hydrolysis were investigated (Table 2 and Figure 4). The Cu^2+^ and Zn^2+^ had an obvious promoting effect on hydrolysis, especially Cu^2+^, which significantly accelerated hydrolysis with the *k* of 0.8071–1.0024 d^−1^, and the promoting ratios were as high as 1570.24–1974.57%. The possible reason is that Cu^2+^ and Zn^2+^ are ions of transition elements, which have strong attraction to ligands and are easy to form complexes, thus promoting the occurrence of hydrolysis. In particular, Cu^2+^ has a strong ability to form complexes. In addition to Cu^2+^, copper in solution can also exist in Cu(OH)^+^, Cu(OH)_3_^−^ and other forms of complexes, which had obvious catalytic effect on hydrolysis and effectively promoted hydrolysis [32].

The hydrolysis was promoted by 0.010 mol L^−1^ of Mn^2+^ with a promotion ratio of 18.75%. With an increase in the concentration of Mn^2+^ (0.050–0.50 mol L^−1^), the hydrolysis was inhibited by Mn^2+^ with the inhibiting ratios ranging from −1.90% to −26.57%. MnCl_2_ × 4H_2_O is a strong acid and weak base salt, and its aqueous solution is weakly acidic, which reduces the pH value of the solution, and resulting in inhibiting the hydrolysis. However, 0.010 mol L^−1^ of Mn^2+^ promoted hydrolysis, possibly because the low concentration of Mn^2+^ did not significantly affect the pH value. The Na^+^, K^+^, Mg^2+^, Ca^2+^, Fe^3+^ and Al^3+^ had inhibiting effect on hydrolysis. The inhibiting effects of Na^+^, K^+^, Mg^2+^ and Ca^2+^ became more pronounced with the increasing concentration, while the inhibiting effect of Al^3+^ was more obvious with the decreasing concentration. The 0.010 mol L^−1^ of Al^3+^ had maximum inhibiting effect with inhibiting ratio of −62.75%. It may be NaCl, KCl and CaCl_2_ are strong acid and base salts that dissolved inorganic salts have a certain salting-out effect on the water solubility of pesticides. The dissolved salt ions (such as Na^+^, K^+^, Ca^2+^, Cl^−^, etc.) in water competed with florpyrauxifen-benzyl molecules for solvent molecules. Moreover, these ions are tightly bound to water in aqueous solution, and may even cause the reduction of the volume of aqueous solution on a macro level, limiting the degree of freedom that water molecules solvated organic molecules. Thus the hydrolysis was prevented. The salting-out effect is more pronounced at higher concentration of dissolved salt [33]. The MgSO_4_, Fe_2_(SO_4_)_3_ × xH_2_O and Al(NO_3_)_3_ × 9H_2_O are strong acid and weak base salts, and their aqueous solutions are weakly acidic, which reduced the pH value in the solutions to a certain extent and inhibited the hydrolysis.

#### 2.2.5. Anions

The natural water environment contains a large number of inorganic salts, mainly cations and anions, which have different effects on hydrolysis of pesticides. Chen et al. reported that nitrate widely existed in water environment obviously promoted hydrolysis of acetochlor, butachlor and metolachlor [41]. The effects of NO_3_^−^ and NO_2_^−^ on hydrolysis of florpyrauxifen-benzyl were investigated (Table 2 and Figure 4). It was found that the presence of NO_3_^−^ and NO_2_^−^ in water had certain effect on hydrolysis with the promoting ratio of 5.03–132.60% and 1.16–48.39%, respectively, and the 50 mg L^−1^ of NO_3_^−^ had the maximum promoting effect with a *T*_0.5_ of only 6.17 d.

#### 2.2.6. Surfactants

The surfactants play an important role in reducing cost by enhancing the emulsification, dispersibility, foaming and wettability of pesticides agents during the processing of pesticides preparations [42]. The surfactants were detected in the environment due to their widespread use and large emissions. The environmental behavior of pesticides and toxicity of other substances in the environment are more likely to be affacted by their special amphipathicity. Yi et al. reported that cationic surfactant of octadecyl trimethyl ammonium bromide (STAB) and nonionic surfactant of nonylphenol polyoxyethylene ether (NPE) promoted the degradation of metolachlor in water-sediment system, while anionic surfactant of SDBS prolonged the degradation *T*_0.5_ of metolachlor. The presence of surfactants affected the environmental behavior of metolachlor [43].

The effects of Tween80, CTAB and SDBS with different micellar concentrations on hydrolysis were investigated (Table 2 and Figure 5). The non-ionic surfactant of Tween80 with low micellar concentration had a slight promoting effect on hydrolysis, with promoting ratios of 4.90–11.18%, probably because non-ionic surfactants are not ionic state in aqueous solution. However, Tween80 with high micellar concentration greatly inhibited hydrolysis, with inhibiting ratios ranging from −31.52% to −60.64%.

The cationic surfactant of CTAB obviously promoted hydrolysis with the *k* of 0.0517–4.2599 d^−1^ and promoting ratio of 8716.06%, and the *T*_0.5_ was shortened to 0.16 d. It may be that the presence of CTAB greatly increased the solubility of florpyrauxifen-benzyl in water, and increased the contacting chance of florpyrauxifen-benzyl and water molecules, thus significantly improving hydrolysis rate.

The hydrolysis of florpyrauxifen-benzyl was inhibited by SDBS with inhibiting ratios of −12.79% to −53.02%. The SDBS is an anionic surfactant, which can increase its solubility by binding with hydrophobic compounds in aqueous solution. Therefore, the solubility of florpyrauxifen-benzyl in water was limited and hydrolysis was inhibited. Meanwhile, steric hindrance may be formed for the long chain of SDBS, which can hinder the chance of florpyrauxifen-benzyl contacting with nucleophile reagents (H_2_O, OH^−^, etc.), thus inhibiting its hydrolysis. In addition, as the micellar concentrations of SDBS increases, the hydrophobic groups of SDBS can self-assemble to generate micelles, forming a non-polar environment in the solution of florpyrauxifen-benzyl and inhibiting its hydrolysis [44].

#### 2.2.7. Coexisting Herbicide of Propyrisulfuron

Sulfonylureas herbicides are extensively used to control weeds of various crops, which exhibit a simple but effective biological mode of action via inhibiting acetolactate synthase, a key enzyme involved in the protein synthesis of plants. Because of the high herbicidal activity of sulfonylureas herbicides, their effective application amounts are low, and sulfonylureas herbicides exhibit extremely low acute and chronic toxicities to mammals in comparison with most herbicides. Hence, the usage of sulfonylurea herbicides is steadily increasing worldwide [29]. As a novel sulfonylurea herbicide, propyrisulfuron has a good control effect on weeds resistant to sulfonylureas herbicides [45]. 

The hydrolysis of florpyrauxifen-benzyl was investigated under the existence of propyrisulfuron (Table 2 and Figure 5). The results showed that propyrisulfuron with different concentrations (1.0–500 mg L^−1^) had a slight promoting effect on hydrolysis, with the promoting ratios ranging from 0.52 to 15.27%. The sulfonylurea herbicides usually hydrolyze faster under acidic condition [46]. It was possible that florpyrauxifen generated from the hydrolysis of florpyrauxifen-benzyl, which rapidly combined with propyrisulfuron, resulting in accelerated hydrolysis of florpyrauxifen-benzyl. In addition, the sulfonyl group and carbonyl group of propyrisulfuron may act as nucleophiles to attack the florpyrauxifen-benzyl molecules, accelerating its hydrolysis.

#### 2.2.8. Humic Acid and Biochar

The humic acid is the main component of organic matter in natural water environment, mainly generated from decomposition products of animals and plants and their by-products. The humic acid has strong chelating ability and its concentration in water is increasing nowadays. Studies have found that hydrolysis rate and degradation pathway of some organic pollutants in water environment were affected by humic acid [30,44]. It was found (Table 2 and Figure 5) that low content of humic acid promoted hydrolysis with promoting ratios of 20.53–45.92% and *T*_0.5_ of 9.83–11.90 d, while high content of humic acid inhibited hydrolysis with an inhibiting ratio of −27.84% and *T*_0.5_ of 19.88 d. The main influencing factors of humic acid on hydrolysis of florpyrauxifen-benzyl were: (1) The humic acid contains a large number of functional groups, such as carboxyl and alcohol hydroxyl groups, which have some strong acid properties that inhibited the activity of hydroxide ions in the solution, thus inhibiting the hydrolysis of florpyrauxifen-benzyl. In addition, humic acid can adsorb pesticides by hydrogen-bond interaction, so hydrolysis was inhibited. (2) The dissolved humic acid can increase the solubility of pesticides, thus promoting hydrolysis of florpyrauxifen-benzyl. The (1) and (2) are two factors in which humic acid has opposite effect on hydrolysis of florpyrauxifen-benzyl. If factor (1) is dominant, hydrolysis will be inhibited. If factor (2) prevails, hydrolysis will be promoted [44]. Dai et al. reported that humic acid inhibited hydrolysis of aldicarb and its oxidized products, aldicarb sulfone and aldicarb sulfoxide, because the factor (1) was dominant [44]. It is concluded that the influence of high content of humic acid on hydrolysis of florpyrauxifen-benzyl was the dominant factor (1), while the influence of low content of humic acid on hydrolysis was the dominant factor (2).

The biochar is a porous material produced by pyrolysis of carbon-rich biomass under anaerobic conditions. It has large specific surface area and abundant sorption sites, and is widely used in ecological restoration, agriculture and environmental protection [47]. The effect of biochar on hydrolysis of florpyrauxifen-benzyl was investigated (Table 2 and Figure 5), indicating that it promoted the hydrolysis with the promoting ratios of 16.08–57.41%. Zhang et al. [48] reported that carbaryl and atrazine hydrolyzed faster in the presence of biochar. On the one hand, the surface sorption of biochar might lead to the aggregation of hydrolytic reactants (compounds or nucleophiles) on the surface of the solid phase, promoting the reaction. On the other hand, biochar contains inorganic ash, which contains inorganic minerals and metal oxides on its surface. The inorganic minerals and metal oxides generated complexes with carbaryl and atrazine to promote their hydrolysis. In addition, the release of metal ions into the solution of biochar also promoted their hydrolysis. It is inferred that biochar promoted the hydrolysis, which might be caused by the sorption of florpyrauxifen-benzyl by biochar, the formation of complexes between inorganic minerals and metal oxides and florpyrauxifen-benzyl, and the release of metal ions from biochar.

### 2.3. Hydrolytic Products and Mechanisms

A large number of studies have shown that pesticides residues in aquatic environment might be transformed into complex metabolites, some metabolites further degraded and became relatively nontoxic substances. However, some metabolites not only degraded slowly, but also their toxicity was even higher than that of the parent compounds, which had potential harmful effects on organisms. According to Glinski et al. and Velisek et al., atrazine metabolites were more toxic than the parent compound and often detected in waters contaminated by pesticides [49,50].

The pesticides may have different degradation products, pathways and mechanisms under different environmental conditions [51]. In order to predict the fate of pesticides in the natural environment and understand the environmental risks that they might pose, it is indispensable for us to understand the chemical reactions and structures of transformation products of pesticides under various environmental conditions [52]. The hydrolytic solutions of florpyrauxifen-benzyl under acidic, neutral and alkaline conditions and the addition of MPs and DFMs were studied by UPLC-QTOF-MS/MS. The total ion chromatograms were shown in Appendix A. It was found that multiple peaks of hydrolytic products appeared after the hydrolysis, and the concentration of florpyrauxifen-benzyl decreased gradually while the concentration of hydrolytic products increased gradually or first increased and then decreased with the extension of time. The possible molecular structures of the hydrolytic products were deduced according to the fragment mass and relative abundance of hydrolytic products and structural characteristics of the parent compound, as shown in Figure 6. The primary and secondary mass spectrogram of each hydrolytic product were shown in Appendix A. The main product of hydrolysis was florpyrauxifen, and degradation mechanism was ester hydrolysis reaction. The florpyrauxifen-benzyl belongs to ester compound and its hydrolysis follows the hydrolysis law of such compounds. When alkaline hydrolysis of ester compounds occurred, most of them belonged to the nucleophilic addition-elimination mechanism. The OH^−^ was a strong nucleophilic reagent, which was directly involved in nucleophilic addition with the carbonyl carbon of ester compounds to form carboxylic acid, and then carboxylic acid neutralized OH^−^ in the solution. Thus, the base disrupted the chemical equilibrium, eliminated the carboxylic acid formed in the reaction and accelerated the hydrolysis rate [12]. It is inferred that the hydrolysis rate of florpyrauxifen-benzyl was faster in alkaline solution, which may be related to its ester bond. In order to ensure the reliability of the results, the retention time and secondary mass spectrograms of florpyrauxifen-benzyl and florpyrauxifen identified in the samples were confirmed and compared with standard substances, and the results showed that the retention time was completely consistent. There were no other new products and pathways in alkaline solutions containing 12 kinds of MPs and 5 parts of DFMs, except some effects on the rate of hydrolytic products formation. In summary, the above products are only proposals for eventual degradation products except that florpyrauxifen has been validated. Moreover, the further confirmation and ecotoxicity etc. of the products will be conducted by synthesis and separation in upcoming study.

Moreover, Miller et al. reported that florpyrauxifen-benzyl and its three metabolites (florpyrauxifen, florpyrauxifen-benzyl hydroxy and florpyrauxifen-hydroxy acid) were determined in natural soil [53]. However, the two metabolites, florpyrauxifen-benzyl hydroxy (C_19_H_12_O_3_N_2_F_2_Cl_2_, [M + H]^+^ = 425.027) and florpyrauxifen-hydroxy acid (C_12_H_6_O_3_N_2_F_2_Cl_2_, [M + H]^+^ = 334.980), determined by Miller et al. were not found in the hydrolytic solutions of florpyrauxifen-benzyl in this experiment, and the total ion chromatogram (TIC) and extraction ion chromatogram (XIC) of the above two products were shown in Appendix A. It is obvious that there is no response intensity other than florpyrauxifen (10.039 min) and florpyrauxifen-benzyl (13.915 min) in the extraction ion chromatograms. It may be that the degradation of pesticides in natural soil is a complex process, which includes sorption, migration, hydrolysis, photolysis, microbial degradation and so on, especially microbial degradation. Moreover, the complex substrates contained in the soil also can affect the degradation process of pesticides. In this experiment, the hydrolytic products of florpyrauxifen-benzyl in ultrapure water after sterilization were identified by an indoor static simulation conditions, and the composition of ultrapure water was relatively simple. 

## 3. Materials and Methods

### 3.1. Instruments and Reagents

Ultra high-performance liquid chromatography (UPLC) (LC-20AD XR) was from Shimadzu (Kyoto, Japan). QTOF-MS/MS (Sciex X500R) was from AB Sciex (Framingham, MA, USA). Analytical balance (XPR26DR/AC) was from Shanghai Mettler Toledo Co., Ltd. (Shanghai, China). Automatic autoclave cooker (BKQ-B50Ⅱ) was from Shandong Boke Biological Industry Co., Ltd. (Jinan, China). Biochemical incubator (SPX150BS-Ⅱ8) was from Shanghai Yiheng Scientific Instrument Co., Ltd. (Shanghai, China). The pH meter (pHS-828) was from Beijing Huarui Boyuan Technology Development Co., Ltd. (Beijing, China). Milli-Q ultrapure water purification system was from Millipore (Burlington, MA, USA).

Chromatographic-grade acetonitrile was obtained from Merck Ltd. (Darmstadt, Germany). Chromatographic-grade formic acid was purchased from Sigma Ltd. (St. Louis, MO, USA). Florpyrauxifen-benzyl (purity ≥ 98.5%) was provided by Dr Ehrenstorfer Ltd. (Augsburg, Germany). Florpyrauxifen (purity ≥ 98.6%) was provided by Alta Scientific Co., Ltd. (Tianjin, China). Analytical-grade NaCl was purchased from Xilong Scientific Co., Ltd. (Shantou, China). Graphitized carbon black (GCB, 60 µm) was bought from Shanghai Anpu Experimental Technology Co., Ltd. (Shanghai, China). Analytical-grade potassium hydrogen phthalate (KHP), NaOH, KH_2_PO_4_, H_3_BO_3_, anhydrous MgSO_4_, KCl, CaCl_2_, Fe_2_(SO_4_)_3_ × xH_2_O, CuSO_4_, MnCl_2_ × 4H_2_O, ZnSO_4_ × 7H_2_O, Al(NO_3_)_3_ × 9H_2_O, NaNO_3_, and NaNO_2_ were purchased from Sinopharm Chemical Reagent Co., Ltd. (Beijing, China). The MPs including polyamide (PA), poly-*β*-hydroxybutyrate (PHB), polyvinyl benzene (PS), polybutanediol succinate (PBS), butylene adipate and butylene terephthalate copolymer (PBAT), low density polyethylene (LDPE), polyhydroxyalkanoates (PHA), polypropylene (PP), polylactice acid (PLA), polymethyl methacrylate (PMMA), polyvinyl chloride (PVC), polyethylene (PE) were from Guangdong Fengtai Plasticizing Co., Ltd. (Dongguan, China). DFMs was purchased from Hubei Meishunhe Medical Technology Co., Ltd. (Wuhan, China), details of the DFMs were supplied in Appendix A. Calcium magnesium phosphate fertilizer (CMPF) was obtained from Yunnan Ruilinfeng Chemical Co., Ltd. (Kunming, China). Urea, potash fertilizer, compound fertilizer were obtained from China Salt Anhui Hongsifang Fertilizer Co., Ltd. (Hefei, China). Organic fertilizer was obtained from Nanchang Jiurun Agricultural Development Co., Ltd. (Nanchang, China). Organic and inorganic compound fertilizer (OICF) was obtained from Jiangxi Yebilv Biochemical Technology Co., Ltd. (Ji’an, China). Propyrisulfuron (purity ≥ 99.8%) was provided by Sumitomo Chemical Co., Ltd. (Tokyo, Japan). Humic acid (purity ≥ 90%) was from Shanghai Macklin Biochemical Technology Co., Ltd. (Shanghai, China). Analytical-grade biochar extracted from plant sclerotia was brought from Tianjin Damao Chemical Reagent Factory (Tianjin, China). Tween80, non-ionic surfactant, was from Tianjin Damao Chemical Reagent Factory (Tianjin, China) and its critical micelle concentration (CMC) was 40 mg L^−1^. Cetyltrimethyl ammonium bromide (CTAB), cationic surfactant, was from Beijing Chemical Factory (Beijing, China) and CMC was 348 mg L^−1^. Sodium dodecylbenzene sulfonate (SDBS), anionic surfactant, was from Shanghai Macklin Biochemical Technology Co., Ltd. (Shanghai, China) and CMC was 550 mg L^−1^. PTFE membrane needle filter (0.22 µm) was obtained from Pall Ltd. (New York, NY, USA). Ultrapure water (pH = 7.12) was produced using a Milli-Q ultrapure water purification system from Millipore (Burlington, MA, USA). Tap water (pH = 7.34) was from laboratory water mains (Nanchang, China). Lake water (pH = 6.54) was from Yaohu lake (Nanchang, China). Paddy water (pH = 7.41) was from experimental field of Jiangxi Agricultural University (Nanchang, China). The lake water and paddy water were filtered with 0.45 mm membranes and stored in the dark at 4 °C until further use. The artificial seawater (pH = 8.18) was prepared according to the references [54,55], as shown in Appendix A, which was almost imitated natural seawater. 

### 3.2. Preparation of Buffer Solutions

Preparation of Clark-Lubs buffer solutions (20 °C) [12]. The buffer solution of pH = 4 was prepared by the steps: 125 mL of 0.1 mol L^−1^ KHP solution and 1.0 mL of 0.1 mol L^−1^ NaOH solution were added to a volumetric bottle, then it was diluted to 250 mL with ultrapure water. 125 mL of 0.1 mol L^−1^ KH_2_PO_4_ solution and 74.08 mL of 0.1 mol L^−1^ NaOH solution were added to a volumetric bottle, and diluted with ultrapure water to obtain the buffer solution of pH = 7. The buffer solution of pH = 9 was prepared as followed. Firstly, 125 mL of 0.1 mol L^−1^ H_3_BO_3_ and 0.1 mol L^−1^ KCl mixed solution was added to a volumetric bottle, then followed by 53.25 mL of 0.1 mol L^−1^ NaOH solution, finally, it was diluted to 250 mL with ultrapure water. Meanwhile, all the pH values of buffer solutions were corrected with 0.1 mol L^−1^ HCl or 0.1 mol L^−1^ NaOH after been sterilized at 121 °C and 0.1 MPa for 30 min.

### 3.3. Hydrolysis Test 

Hydrolysis test was conducted according to “Test guidelines on environmental safety assessment for chemical pesticides” and references [56,57,58].

(1)An amount of 1 mg L^−1^ of florpyrauxifen-benzyl aqueous solutions was prepared, and its hydrolysis was conducted at different temperatures (15, 25, 35, 50 °C) and pH values (4, 7, 9), respectively. (2)The initial mass concentrations of 1, 2 and 5 mg L^−1^ of florpyrauxifen-benzyl were prepared with the pH = 7 buffer solutions, and its hydrolysis was conducted at 25 °C, respectively. (3)An amount of 1 mg L^−1^ of florpyrauxifen-benzyl aqueous solutions was prepared with ultrapure water, tap water, lake water, paddy water and artificial seawater, and its hydrolysis was conducted at 25 °C, respectively. (4)An amount of 7 mg L^−1^ of florpyrauxifen-benzyl was prepared with the pH = 9 buffer solutions, and the effects of different environmental factors on hydrolysis were conducted at 35 °C, respectively, including 12 kinds of common MPs (PA, PHB, PS, PBS, PBAT, LDPE, PHA, PP, PLA, PMMA, PVC, PE) with different contents (0.050, 0.10, 0.25, 0.50%), different contents (0.050, 0.10, 0.25, 0.50%) of DFMs which were divided into the whole mask, outer layer, middle layer, inner layer and ear band and they were respectively cut into tiny pieces, 6 kinds of common fertilizers (CMPF, urea, organic fertilizer, potash fertilizer, compound fertilizer, OICF) with different contents (0.050, 0.10, 0.25, 0.50%), 9 kinds of cations (Na^+^, K^+^, Mg^2+^, Ca^2+^, Fe^3+^, Cu^2+^, Mn^2+^, Zn^2+^, Al^3+^) with different concentrations (0.010, 0.050, 0.10, 0.50 mol L^−1^), 2 kinds of anions with different concentrations (0.10, 1.0, 10, 50 mg L^−1^ of NO_3_^−^ and 0.010, 0.10, 1.0, 10 mg L^−1^ of NO_2_^−^), 3 kinds of surfactants with different critical micelle concentrations (1.0, 2.0, 5.0, 50 CMC of Tween80, 0.10, 0.50, 1.0, 10 CMC of CTAB and 0.10, 0.50, 1.0, 10 CMC of SDBS), coexisting herbicide propyrisulfuron (1.0, 10, 100, 500 mg L^−1^), humic acid and biochar with different contents (0.050, 0.10, 0.25, 0.50%), as well as the blank control group.

The brown glass bottles were selected and sealed with a sealing film to avoid the influence of photolysis, volatilization and oxidation on hydrolysis during culture and sampling. All samples were placed in a constant temperature incubator away from light. The samples were collected for more than 7 times according to degradation conditions, and the content of florpyrauxifen-benzyl was determined until the test was terminated at 120 days. Three replicates were conducted for per treatment. 

### 3.4. Sample Analysis Method

#### 3.4.1. Preparation of Standard and Matrix-Matched Standard Working Solutions

The standard stock solution of florpyrauxifen-benzyl (100 mg L^−1^) was prepared in acetonitrile. The standard stock solution was gradually diluted with acetonitrile to prepare standard working solutions with mass concentrations of 5, 10, 25, 50, 200, 500 and 1000 µg L^−1^. In addition the matrix-matched standard working solutions (5, 10, 25, 50, 200, 500 and 1000 µg L^−1^) were prepared by diluting the standard stock solution with different blank water matrix solutions, respectively. All solutions were stored at 4 °C and protected from light.

#### 3.4.2. Sample Pretreatment

An amount of 5 mL of water sample was accurately removed into a 50 mL centrifuge tube, followed by the addition of 20 mL of acetonitrile to extract florpyrauxifen-benzyl under vortex for 5 min. Then, 2 g of NaCl and 2 g of anhydrous MgSO_4_ were added under vortex for 1 min [59]. The extract was centrifuged at 9000 rpm min^−1^ for 5 min and the aqueous and organic phases were stratified. After extraction, 1.5 mL of supernatant was transferred to a 2.0 mL centrifuge tube containing 5 mg of GCB and 150 mg of anhydrous MgSO_4_. Then the centrifuge tube was shaken using vortex for 1 min and centrifuged at 12000 rpm min^−1^ for 10 min. The purified supernatant was filtered through 0.22 µm filter membrane for analysis [27].

#### 3.4.3. Liquid Chromatography and Mass Spectrometry Conditions

The florpyrauxifen-benzyl was determined by UPLC-QTOF-MS/MS with a Waters CORTECSTM UPLC C18 column (100 × 2.1 mm, 1.6 µm, Milford, MA, USA) at 40 °C. The mobile phase was acetonitrile and 0.1% of formic acid aqueous solution (60/40, V/V). The flow rate, injection volume and autosampler temperature were 0.3 mL min^−1^, 4 μL and 4 °C, respectively [27]. The detailed parameters of QTOF-MS/MS were shown in Appendix A. 

### 3.5. Data Analysis 

The hydrolysis law of florpyrauxifen-benzyl is described by a first-order kinetics model, and the kinetics parameters of hydrolysis are obtained by nonlinear fitting method (Formula (1)). The half-life (*T*_0.5_) is calculated by Formula (2). The temperature effect coefficient (*Q*) is defined as the ratio of hydrolysis rate constant *k*_t_ at a certain temperature to *k*_t+10_ when temperature is higher than 10 °C (Formula (3)). According to Arrhenius formula (Formula (4)) of the relation between temperature and hydrolysis rate constant, logarithm is taken from both sides of Formula (4) to obtain Formula (5). Plotting ln*k* vs *T*^−1^ give a straight line with a slope equal to −*E*_a_/*R* with an intercept of ln*A*, which can be seen that ln*k* and *T*^−1^ have a linear relationship. Hence, the activation energy (*E*_a_) can be obtained which refers to the energy difference between the excited state and ground state of the reactant molecule. The activation enthalpy (Δ*H*) is calculated by the *E*_a_ (Formula (6)). It is related to the energy barrier of compound reaction, which is used to judge the degree of bond breaking. The activation entropy (Δ*S*) is used to judge the degree of system disorder, reflecting the gain and loss of freedom between the initial compound and the transition state (activated complex), which is calculated by Formula (7). The Gibbs free energy (Δ*G*) is calculated by Formula (8) [60]. The promoting or inhibiting effect of environmental factor on hydrolysis is calculated by Formula (9) [61].
(1)$$C_{t}=C_{0}\times \exp^{-kt}$$
(2)$$T_{0.5}=\ln2/k$$
(3)$$Q=\frac{k_{t+10}}{k_{t}}$$
(4)$$k=A\exp^{-Ea/RT}$$
(5)lnk=lnA−Ea/RT
(6)$$\Delta H=E_{a}-RT$$
(7)$$\Delta S=R\;(\ln A-\ln\frac{k_{B}T}{h})$$
(8)ΔG=ΔH−TΔS
(9)PR/IR(%)=(ki−k0)/k0×100%
where, *C*_t_ is the mass concentration of florpyrauxifen-benzyl (μg L^−1^) at a given time of *t*, *C*_0_ is the initial mass concentration of florpyrauxifen-benzyl (μg L^−1^), *k* is the hydrolysis rate constant (d^−1^), *t* is the reaction time (d), *T*_0.5_ is the hydrolysis half-life (d), *Q* is the temperature effect coefficient, *A* is the preexponential factor, *E*_a_ is the activation energy (J mol^−1^), *R* is the molar gas constant [8.314 J (mol K)^−1^], *T* is the thermodynamic temperature (K), Δ*H* is the activation enthalpy (J mol^−1^), Δ*S* is the activation entropy [J (mol K)^−1^], *k*_B_ is the Boltzmann constant (1.381 × 10^−23^ J K^−1^), *h* is Planck constant (6.626 × 10^−34^ J s), Δ*G* is the Gibbs free energy (J mol^−1^), *PR/IR* is the promoting or inhibiting ratio (%), *k*_0_ is the hydrolysis rate constant without other factors, *k*_i_ is the hydrolysis rate constant with the presence of environmental factors, and a positive value represents the promoting effect while a negative value represents the inhibiting effect.

### 3.6. Hydrolytic Mechanism Study 

An amount of 5 mg L^−1^ of the florpyrauxifen-benzyl aqueous solutions was prepared with the pH = 4, 7, 9 buffer solutions, respectively. An amount of 5 mg L^−1^ of the florpyrauxifen-benzyl aqueous solutions was prepared with the pH = 9 buffer solutions, and 0.25% of 12 kinds of MPs and the whole mask, outer layer, middle layer, inner layer and ear band of DFMs were added respectively. All solutions were cultured in a constant temperature incubator at 50 °C. The samples were collected at 0, 1, 2, 3, 4 and 6 days and treated according to Section 3.4. The possible hydrolysis mechanisms were proposed according to the degradation products. 

## 4. Conclusions

In this study, the hydrolytic behavior, various influencing factors and mechanisms of florpyrauxifen-benzyl were systematically investigated. It was found that the hydrolysis reaction of florpyrauxifen-benzyl was a spontaneous process followed the first-order kinetics, and it was driven by endothermic, base catalysis and activation entropy increase. The hydrolysis rate was affected by many factors, among which pH value, temperature, butylene adipate and butylene terephthalate copolymer (PBAT), Cu^2+^ and cetyltrimethyl ammonium bromide (CTAB) had significant influence. The degradation of florpyrauxifen-benzyl was slow in acidic conditions and at low temperatures. Therefore, when using florpyrauxifen-benzyl in farmlands, the variations in the physical and chemical indexes in different regions should be considered to prevent the pollution of surface water. Based on the identification of hydrolytic products, several possible hydrolytic pathways were proposed. The main product was florpyrauxifen, and the degradation mechanism was ester hydrolysis reaction. The addition of microplastics (MPs) and disposable face masks (DFMs) had no effect on the hydrolytic products and pathways except some effect on the rate of hydrolytis. The results can not only predict the residual characteristics of florpyrauxifen-benzyl in aqueous environment and the mechanisms of its migration and transformation, but also provide a scientific basis for the evaluating the impact of florpyrauxifen-benzyl on the ecosystem. 

## Figures and Tables

**Figure 1 ijms-24-10521-f001:**
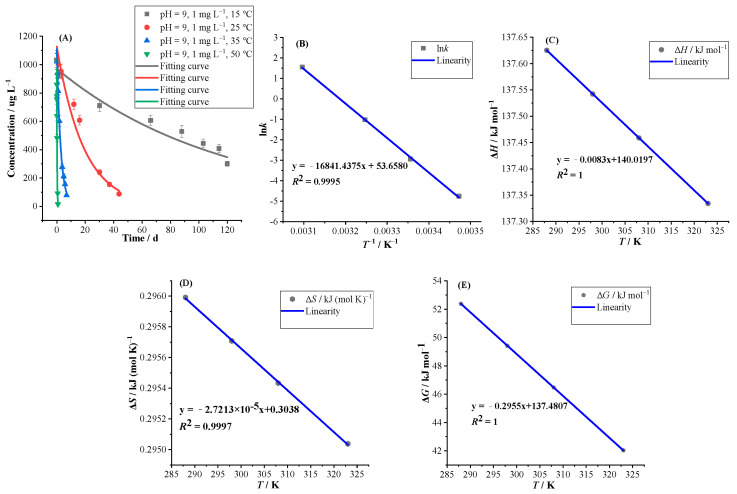
(**A**) Hydrolytic curves of florpyrauxifen-benzyl in pH = 9 buffer solutions at different temperatures (n = 3) by first-order kinetics model; (**B**) relationship between hydrolysis rate constant (ln*k*) and temperature (*T*^−1^) in pH = 9 buffer solutions; (**C**–**E**) relationship between activation enthalpy (Δ*H*), activation entropy (Δ*S*) and Gibbs free energy (Δ*G*) and temperature (*T*) in pH = 9 buffer solutions.

**Figure 2 ijms-24-10521-f002:**
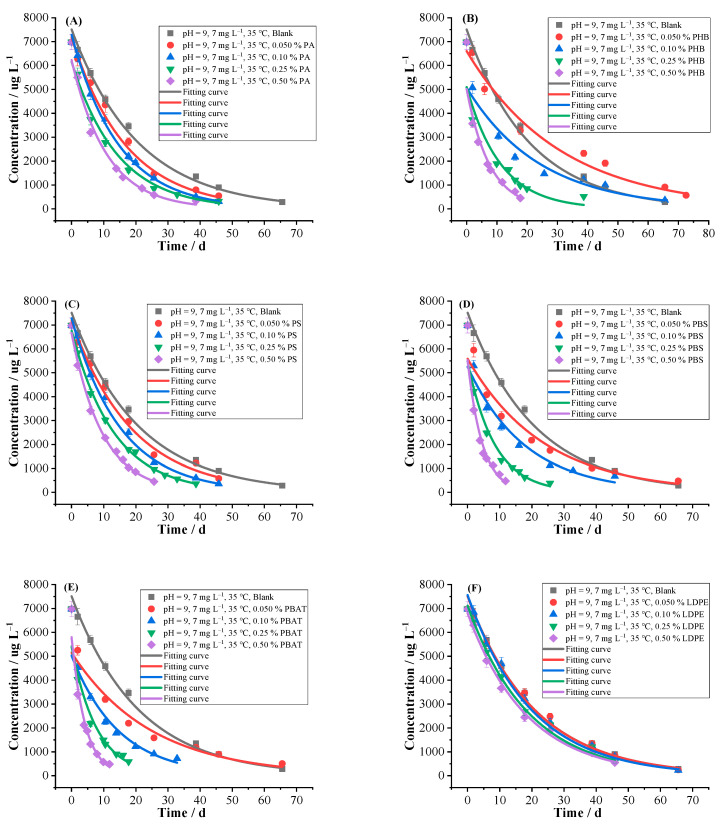
Hydrolytic curves of 7 mg L ^−1^ of florpyrauxifen-benzyl under different influencing factor (**A**) PA, (**B**) PHB, (**C**) PS, (**D**) PBS, (**E**) PBAT, (**F**) LDPE, (**G**) PHA, (**H**) PP, (**I**) PLA, (**J**) PMMA, (**K**) PVC and (**L**) PE in pH = 9 buffer solutions at 35 °C (n = 3) by first-order kinetics model.

**Figure 3 ijms-24-10521-f003:**
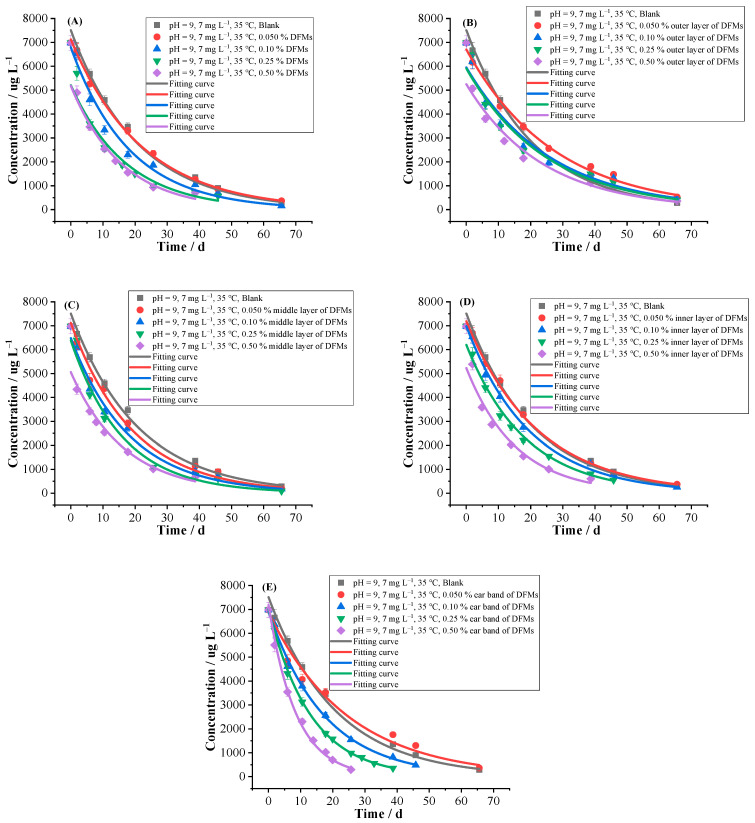
Hydrolytic curves of 7 mg L^−1^ florpyrauxifen-benzyl under different influencing factor (**A**) DFMs, (**B**) outer layer of DFMs, (**C**) middle layer of DFMs, (**D**) inner layer of DFMs and (**E**) ear band of DFMs in pH = 9 buffer solutions at 35 °C (n = 3) by first-order kinetics model.

**Figure 4 ijms-24-10521-f004:**
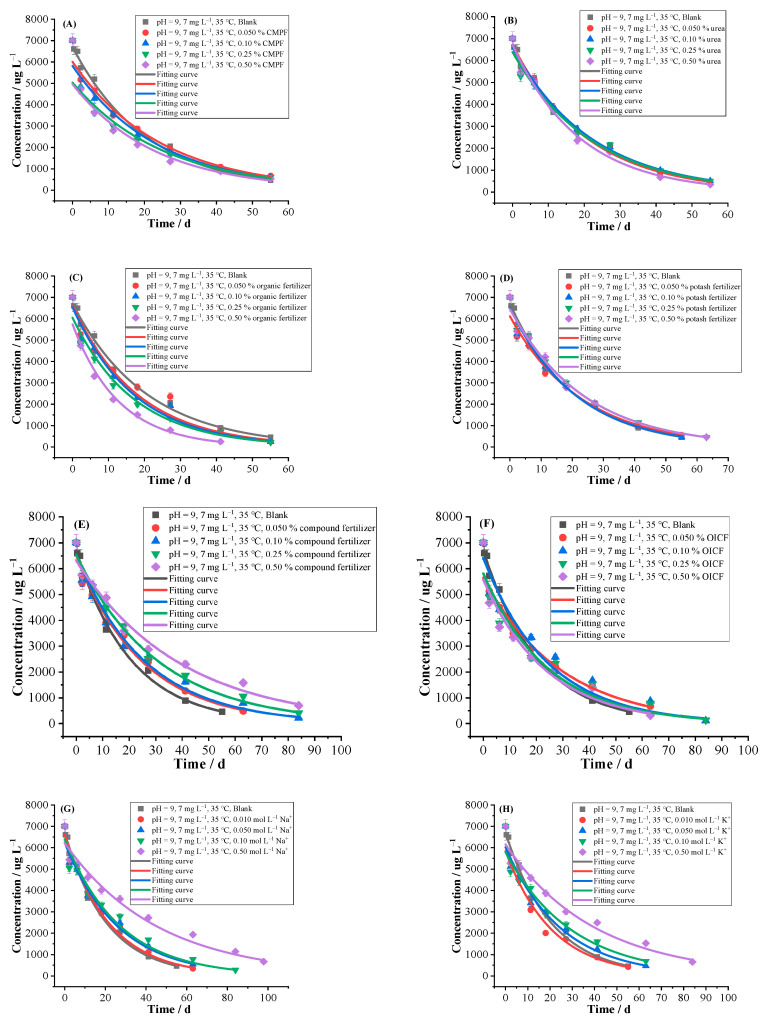
Hydrolytic curves of 7 mg L^−1^ of florpyrauxifen-benzyl under different influencing factor (**A**) CMPF, (**B**) urea, (**C**) organic fertilizer, (**D**) potash fertilizer, (**E**) compound fertilizer, (**F**) OICF, (**G**) Na^+^, (**H**) K^+^, (**I**) Mg^2+^,(**J**) Ca^2+^, (**K**) Fe^3+^,(**L**) Cu^2+^, (**M**) Mn^2+^, (**N**) Zn^2+^, (**O**) Al^3+^, (**P**) NO_3_^−^ and (**Q**) NO_2_^−^ in pH = 9 buffer solutions at 35 °C (n = 3) by first-order kinetics model.

**Figure 5 ijms-24-10521-f005:**
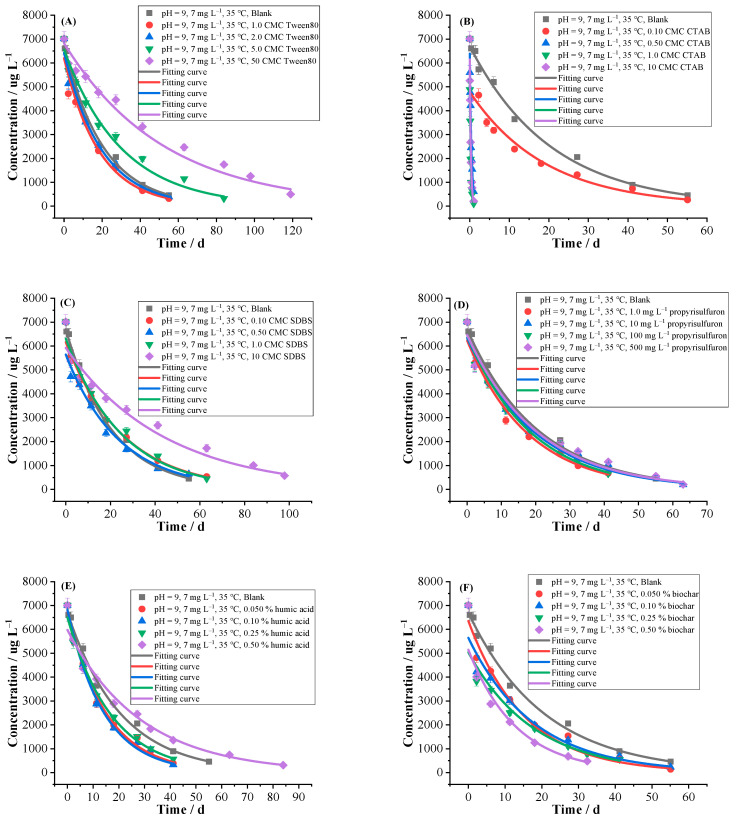
Hydrolytic curves of 7 mg L^−1^ of florpyrauxifen-benzyl under different influencing factor (**A**) Tween80, (**B**) CTAB, (**C**) SDBS, (**D**) propyrisulfuron, (**E**) humic acid and (**F**) biochar in pH = 9 buffer solutions at 35 °C (n = 3) by first-order kinetics model.

**Figure 6 ijms-24-10521-f006:**
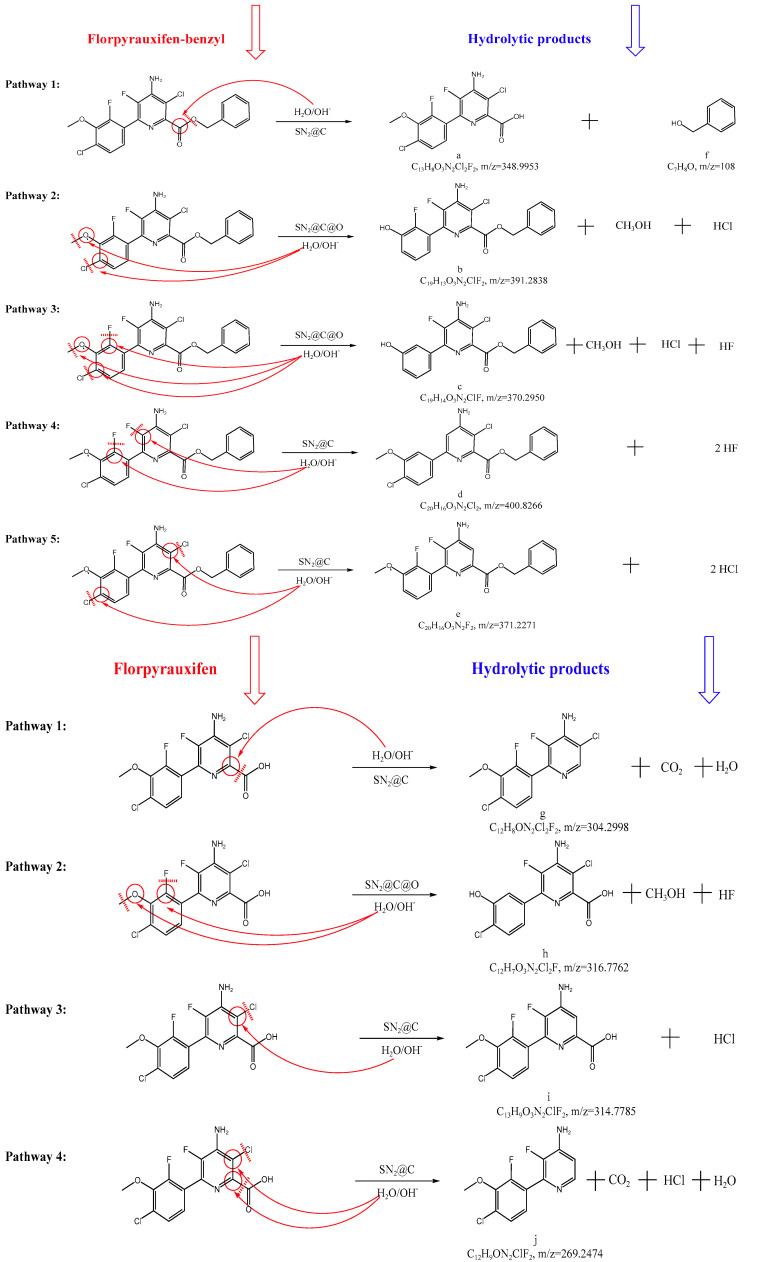
Proposed hydrolysis pathways of florpyrauxifen-benzyl in aqueous solution. Where SN_2_@C and SN_2_@C@O represented bimolecular nucleophilic substitution mechanism of nucleophiles (H_2_O, OH^−^, etc.) attack C atom or C and O atoms, respectively. The primary and secondary mass spectrograms of the hydrolytic products (a–j) were shown in Appendix A.

**Table 1 ijms-24-10521-t001:** Hydrolytic kinetics parameters of florpyrauxifen-benzyl solutions at different temperatures, pH values, initial mass concentrations and water types (n = 3).

Water	Mass Concentration/mg L^−1^	pH	Temperature/°C	Kinetic Equation	*R*2	Rate Constant *(k)*/d^−1^	Half-Life (*T*0.5)/d
Ultrapure water	1	4	15	*C*_t_ = 1021.11e^−0.0042t^	0.9772	0.0042	163.48
			25	*C*_t_ = 960.23e^−0.0031t^	0.9537	0.0031	220.75
			35	*C*_t_ = 972.30e^−0.0034t^	0.9630	0.0034	202.67
			50	*C*_t_ = 994.75e^−0.0034t^	0.8538	0.0034	205.68
		7	15	*C*_t_ = 1011.04e^−0.0052t^	0.9429	0.0052	134.59
			25	*C*_t_ = 1154.46e^−0.0409t^	0.9708	0.0409	16.96
			35	*C*_t_ = 1194.53e^−0.0506t^	0.9250	0.0506	13.70
			50	*C*_t_ = 1199.24e^−0.0702t^	0.9154	0.0702	9.87
		9	15	*C*_t_ = 970.39e^−0.0086t^	0.9403	0.0086	80.88
			25	*C*_t_ = 1128.19e^−0.0529t^	0.9706	0.0529	13.10
			35	*C*_t_ = 1111.91e^−0.3573t^	0.9875	0.3573	1.94
			50	*C*_t_ = 989.59e^−4.6981t^	0.9850	4.6981	0.15
Ultrapure water	1	7	25	*C*_t_ = 1154.46e^−0.0409t^	0.9708	0.0409	16.96
	2			*C*_t_ = 2286.14e^−0.0238t^	0.9507	0.0238	29.10
	5			*C*_t_ = 5016.71e^−0.0039t^	0.9717	0.0039	176.82
Ultrapure water	1	7.12	25	*C*_t_ = 1154.46e^−0.0409t^	0.9708	0.0409	16.96
Tap water		7.34		*C*_t_ = 1056.08e^−0.0144t^	0.9278	0.0144	48.04
Lake water		6.54		*C*_t_ = 1150.18e^−0.0287t^	0.9541	0.0287	24.13
Paddy water		7.41		*C*_t_ = 1159.70e^−0.0238t^	0.9551	0.0238	29.11
Seawater		8.18		*C*_t_ = 969.31e^−0.0270t^	0.8850	0.0270	25.66

**Table 2 ijms-24-10521-t002:** Hydrolytic kinetics parameters of 7 mg L^−1^ florpyrauxifen-benzyl under different influencing factors in pH = 9 buffer solutions at 35 °C (n = 3).

Influencing Factors		Kinetic Equation	*R* ^2^	Rate Constant *(k)*/d^−1^	Half-Life (*T*0.5)/d	Promoting or Inhibiting Ratio *(PR/IR)*/%
Blank control group 1		*C*_t_ = 7510.60e^−0.0480*t*^	0.9933	0.0480	14.43	/
^1^ PA/%	0.050	*C*_t_ = 7177.88e^−0.0589*t*^	0.9893	0.0589	11.78	22.57
	0.10	*C*_t_ = 7285.07e^−0.0681*t*^	0.9984	0.0681	10.18	41.84
	0.25	*C*_t_ = 6158.84e^−0.0700*t*^	0.9780	0.0700	9.91	45.67
	0.50	*C*_t_ = 6229.83e^−0.0914*t*^	0.9669	0.0914	7.59	90.25
^1^ PHB/%	0.050	*C*_t_ = 6602.58e^−0.0321*t*^	0.9718	0.0321	21.57	−33.07
	0.10	*C*_t_ = 5067.25e^−0.0409*t*^	0.9280	0.0409	16.95	−14.85
	0.25	*C*_t_ = 5100.17e^−0.0888*t*^	0.8597	0.0888	7.81	84.90
	0.50	*C*_t_ = 5014.89e^−0.1317*t*^	0.9503	0.1317	5.26	174.22
^1^ PS/%	0.050	*C*_t_ = 7179.06e^−0.0532*t*^	0.9800	0.0532	13.03	10.77
	0.10	*C*_t_ = 7283.69e^−0.0654*t*^	0.9958	0.0654	10.59	36.28
	0.25	*C*_t_ = 6760.08e^−0.0755*t*^	0.9967	0.0755	9.18	57.27
	0.50	*C*_t_ = 6670.89e^−0.1036*t*^	0.9966	0.1036	6.69	115.76
^1^ PBS/%	0.050	*C*_t_ = 5585.40e^−0.0420*t*^	0.9602	0.0420	16.50	−12.49
	0.10	*C*_t_ = 5248.99e^−0.0553*t*^	0.9187	0.0553	12.54	15.10
	0.25	*C*_t_ = 5468.54e^−0.1190*t*^	0.9550	0.1190	5.83	147.75
	0.50	*C*_t_ = 5528.74e^−0.2173*t*^	0.9484	0.2173	3.19	352.42
^1^ PBAT/%	0.050	*C*_t_ = 5148.41e^−0.0403*t*^	0.9436	0.0403	17.20	−16.06
	0.10	*C*_t_ = 5029.84e^−0.0676*t*^	0.9391	0.0676	10.25	40.84
	0.25	*C*_t_ = 5413.17e^−0.1281*t*^	0.9543	0.1281	5.41	166.83
	0.50	*C*_t_ = 5804.29e^−0.2334*t*^	0.9642	0.2334	2.97	385.99
^1^ LDPE/%	0.050	*C*_t_ = 7504.82e^−0.0481*t*^	0.9853	0.0481	14.41	0.19
	0.10	*C*_t_ = 7561.63e^−0.0508*t*^	0.9897	0.0508	13.65	5.77
	0.25	*C*_t_ = 7118.21e^−0.0526*t*^	0.9850	0.0526	13.18	9.50
	0.50	*C*_t_ = 6797.73e^−0.0538*t*^	0.9948	0.0538	12.89	11.97
^1^ PHA/%	0.050	*C*_t_ = 6541.85e^−0.0320*t*^	0.9924	0.0320	21.69	−33.47
	0.10	*C*_t_ = 5442.54e^−0.0424*t*^	0.9355	0.0424	16.34	−11.66
	0.25	*C*_t_ = 5193.77e^−0.1115*t*^	0.9167	0.1115	6.22	132.15
	0.50	*C*_t_ = 5280.78e^−0.1490*t*^	0.9519	0.1490	4.65	210.37
^1^ PP/%	0.050	*C*_t_ = 7633.40e^−0.0512*t*^	0.9893	0.0512	13.54	6.58
	0.10	*C*_t_ = 7420.42e^−0.0551*t*^	0.9958	0.0551	12.58	14.72
	0.25	*C*_t_ = 7422.28e^−0.0598*t*^	0.9943	0.0598	11.59	24.51
	0.50	*C*_t_ = 7318.79e^−0.0649*t*^	0.9911	0.0649	10.68	35.15
^1^ PLA/%	0.050	*C*_t_ = 6795.70e^−0.0364*t*^	0.9952	0.0364	19.06	−24.26
	0.10	*C*_t_ = 6657.12e^−0.0412*t*^	0.9929	0.0412	16.80	−14.10
	0.25	*C*_t_ = 5643.14e^−0.0476*t*^	0.9694	0.0476	14.56	−0.85
	0.50	*C*_t_ = 5300.35e^−0.0485*t*^	0.9528	0.0485	14.29	1.04
^1^ PMMA/%	0.050	*C*_t_ = 7072.02e^−0.0762*t*^	0.9986	0.0762	9.10	58.66
	0.10	*C*_t_ = 6569.37e^−0.0883*t*^	0.9955	0.0883	7.85	83.92
	0.25	*C*_t_ = 6930.55e^−0.1323*t*^	0.9867	0.1323	5.24	175.57
	0.50	*C*_t_ = 7023.22e^−0.1895*t*^	0.9902	0.1895	3.66	294.65
^1^ PVC/%	0.050	*C*_t_ = 7348.99e^−0.0473*t*^	0.9946	0.0473	14.66	−1.52
	0.10	*C*_t_ = 7212.53e^−0.0478*t*^	0.9916	0.0478	14.49	−0.37
	0.25	*C*_t_ = 6991.86e^−0.0537*t*^	0.9982	0.0537	12.91	11.81
	0.50	*C*_t_ = 6633.69e^−0.0553*t*^	0.9971	0.0553	12.53	15.22
^1^ PE/%	0.050	*C*_t_ = 7319.65e^−0.0423*t*^	0.9914	0.0423	16.37	−11.83
	0.10	*C*_t_ = 7418.69e^−0.0500*t*^	0.9907	0.0500	13.86	4.16
	0.25	*C*_t_ = 7104.14e^−0.0508*t*^	0.9898	0.0508	13.64	5.83
	0.50	*C*_t_ = 6745.45e^−0.0512*t*^	0.9872	0.0512	13.52	6.73
^1^ DFMs/%	0.050	*C*_t_ = 7126.03e^−0.0451*t*^	0.9984	0.0451	15.38	−6.16
	0.10	*C*_t_ = 6795.57e^−0.0547*t*^	0.9808	0.0547	12.67	13.95
	0.25	*C*_t_ = 5208.87e^−0.0576*t*^	0.9160	0.0576	12.03	19.97
	0.50	*C*_t_ = 5184.28e^−0.0631*t*^	0.9166	0.0631	10.98	31.42
^1^ Outer layer of DFMs/%	0.050	*C*_t_ = 6702.83e^−0.0365*t*^	0.9904	0.0365	18.97	−23.91
	0.10	*C*_t_ = 5953.48e^−0.0384*t*^	0.9635	0.0384	18.04	−19.97
	0.25	*C*_t_ = 5906.58e^−0.0399*t*^	0.9627	0.0399	17.38	−16.95
	0.50	*C*_t_ = 5248.56e^−0.0424*t*^	0.9756	0.0424	16.36	−11.79
^1^ Middle layer of DFMs/%	0.050	*C*_t_ = 7123.12e^−0.0521*t*^	0.9860	0.0521	13.31	8.43
	0.10	*C*_t_ = 6481.00e^−0.0544*t*^	0.9899	0.0544	12.75	13.20
	0.25	*C*_t_ = 6387.03e^−0.0622*t*^	0.9649	0.0622	11.15	29.47
	0.50	*C*_t_ = 5060.24e^−0.0600*t*^	0.9532	0.0600	11.55	24.97
^1^ Inner layer of DFMs/%	0.050	*C*_t_ = 7189.70e^−0.0453*t*^	0.9983	0.0453	15.29	−5.62
	0.10	*C*_t_ = 6981.13e^−0.0504*t*^	0.9932	0.0504	13.75	4.96
	0.25	*C*_t_ = 6198.20e^−0.0542*t*^	0.9909	0.0542	12.79	12.85
	0.50	*C*_t_ = 5230.34e^−0.0648*t*^	0.9538	0.0648	10.70	34.84
^1^ Ear band of DFMs/%	0.050	*C*_t_ = 6874.53e^−0.0405*t*^	0.9624	0.0405	17.12	−15.70
	0.10	*C*_t_ = 6913.16e^−0.0574*t*^	0.9973	0.0574	12.08	19.49
	0.25	*C*_t_ = 7009.87e^−0.0763*t*^	0.9987	0.0763	9.09	58.83
	0.50	*C*_t_ = 7189.25e^−0.1168*t*^	0.9894	0.1168	5.93	143.21
Blank control group 2		*C*_t_ = 6703.39e^−0.0483*t*^	0.9948	0.0483	14.34	/
^2^ CMPF/%	0.050	*C*_t_ = 6005.36e^−0.0409*t*^	0.9885	0.0409	16.95	−15.36
	0.10	*C*_t_ = 5816.78e^−0.0424*t*^	0.9853	0.0424	16.34	−12.19
	0.25	*C*_t_ = 5036.41e^−0.0398*t*^	0.9658	0.0398	17.40	−17.57
	0.50	*C*_t_ = 4961.20e^−0.0435*t*^	0.9604	0.0435	15.92	−9.89
^2^ Urea/%	0.050	*C*_t_ = 6725.00e^−0.0483*t*^	0.9898	0.0483	14.36	−0.08
	0.10	*C*_t_ = 6597.26e^−0.0453*t*^	0.9949	0.0453	15.30	−6.25
	0.25	*C*_t_ = 6387.59e^−0.0453*t*^	0.9900	0.0453	15.29	−6.21
	0.50	*C*_t_ = 6717.64e^−0.0535*t*^	0.9912	0.0535	12.97	10.64
^2^ Organic fertilizer/%	0.050	*C*_t_ = 6616.29e^−0.0552*t*^	0.9698	0.0552	12.57	14.16
	0.10	*C*_t_ = 6539.83e^−0.0576*t*^	0.9775	0.0576	12.04	19.18
	0.25	*C*_t_ = 6051.00e^−0.0582*t*^	0.9829	0.0582	11.91	20.49
	0.50	*C*_t_ = 5739.93e^−0.0773*t*^	0.9852	0.0773	8.96	60.06
^2^ Potash fertilizer/%	0.050	*C*_t_ = 6115.86e^−0.0435*t*^	0.9909	0.0435	15.92	−9.89
	0.10	*C*_t_ = 6516.86e^−0.0471*t*^	0.9936	0.0471	14.72	−2.52
	0.25	*C*_t_ = 6475.41e^−0.0421*t*^	0.9972	0.0421	16.46	−12.83
	0.50	*C*_t_ = 6454.15e^−0.0421*t*^	0.9943	0.0421	16.46	−12.85
^2^ Compound fertilizer/%	0.050	*C*_t_ = 6513.16e^−0.0399*t*^	0.9920	0.0399	17.36	−17.38
	0.10	*C*_t_ = 6551.27e^−0.0386*t*^	0.9559	0.0386	17.95	−20.07
	0.25	*C*_t_ = 6528.45e^−0.0322*t*^	0.9918	0.0322	21.56	−33.46
	0.50	*C*_t_ = 6306.80e^−0.0254*t*^	0.9826	0.0254	27.32	−47.50
^2^ OICF/%	0.050	*C*_t_ = 5650.74e^−0.0343*t*^	0.9836	0.0343	20.23	−29.10
	0.10	*C*_t_ = 6418.24e^−0.0413*t*^	0.9185	0.0413	16.80	−14.61
	0.25	*C*_t_ = 5815.25e^−0.0409*t*^	0.9500	0.0409	16.96	−15.40
	0.50	*C*_t_ = 5575.26e^−0.0420*t*^	0.9557	0.0420	16.50	−13.06
^2^ Na^+^/mol ^L−1^	0.010	*C*_t_ = 6603.34e^−0.0452*t*^	0.9870	0.0452	15.32	−6.37
	0.050	*C*_t_ = 6237.02e^−0.0380*t*^	0.9905	0.0380	18.23	−21.32
	0.10	*C*_t_ = 6392.31e^−0.0370*t*^	0.9796	0.0370	18.73	−23.43
	0.50	*C*_t_ = 6215.34e^−0.0218*t*^	0.9798	0.0218	31.87	−54.99
^2^ K^+^/mol ^L−1^	0.010	*C*_t_ = 5848.30e^−0.0480*t*^	0.9741	0.0480	14.45	−0.75
	0.050	*C*_t_ = 6020.21e^−0.0400*t*^	0.9877	0.0400	17.32	−17.20
	0.10	*C*_t_ = 5786.43e^−0.0334*t*^	0.9790	0.0334	20.72	−30.77
	0.50	*C*_t_ = 6154.19e^−0.0250*t*^	0.9795	0.0250	27.76	−48.32
^2^ Mg^2+^/mol ^L−1^	0.010	*C*_t_ = 6003.34e^−0.0425*t*^	0.9888	0.0425	16.32	−12.11
	0.050	*C*_t_ = 5865.90e^−0.0354*t*^	0.9821	0.0354	19.56	−26.68
	0.10	*C*_t_ = 5690.02e^−0.0321*t*^	0.9728	0.0321	21.62	−33.65
	0.50	*C*_t_ = 4007.15e^−0.0311*t*^	0.9425	0.0311	22.32	−35.72
^2^ Ca^2+^/mol ^L−1^	0.010	*C*_t_ = 5552.82e^−0.0371*t*^	0.9783	0.0371	18.70	−23.30
	0.050	*C*_t_ = 5358.55e^−0.0287*t*^	0.9684	0.0287	24.17	−40.65
	0.10	*C*_t_ = 5508.63e^−0.0327*t*^	0.9518	0.0327	21.18	−32.28
	0.50	*C*_t_ = 5008.52e^−0.0236*t*^	0.9615	0.0236	29.36	−51.14
^2^ Fe^3+^/mol ^L−1^	0.010	*C*_t_ = 5334.64e^−0.0240*t*^	0.9660	0.0240	28.86	−50.29
	0.050	*C*_t_ = 4038.75e^−0.0343*t*^	0.8630	0.0343	20.18	−28.93
	0.10	*C*_t_ = 4166.93e^−0.0434*t*^	0.9291	0.0434	15.99	−10.29
	0.50	*C*_t_ = 6300.30e^−0.0220*t*^	0.9820	0.0220	31.46	−54.41
^2^ Cu^2+^/mol ^L−1^	0.010	*C*_t_ = 8057.62e^−0.8071*t*^	0.9662	0.8071	0.86	1570.24
	0.050	*C*_t_ = 7394.43e^−0.9257*t*^	0.9870	0.9257	0.75	1815.79
	0.10	*C*_t_ = 5852.22e^−1.0024*t*^	0.9558	1.0024	0.69	1974.57
	0.50	*C*_t_ = 5642.79e^−0.9059*t*^	0.9655	0.9059	0.77	1774.79
^2^ Mn^2+^/mol ^L−1^	0.010	*C*_t_ = 5309.73e^−0.0574*t*^	0.9643	0.0574	12.08	18.75
	0.050	*C*_t_ = 5664.72e^−0.0474*t*^	0.9635	0.0474	14.62	−1.90
	0.10	*C*_t_ = 5456.66e^−0.0455*t*^	0.9555	0.0455	15.23	−5.79
	0.50	*C*_t_ = 5522.17e^−0.0355*t*^	0.9584	0.0355	19.54	−26.57
^2^ Zn^2+^/mol ^L−1^	0.010	*C*_t_ = 6230.32e^−0.0762*t*^	0.9593	0.0762	9.09	57.80
	0.050	*C*_t_ = 4460.53e^−0.0886*t*^	0.9478	0.0886	7.82	83.44
	0.10	*C*_t_ = 4785.48e^−0.0947*t*^	0.9599	0.0947	7.32	95.99
	0.50	*C*_t_ = 6544.89e^−0.0748*t*^	0.9168	0.0748	9.27	54.70
^2^ Al^3+^/mol ^L−1^	0.010	*C*_t_ = 6369.25e^−0.0180*t*^	0.9578	0.0180	38.51	−62.75
	0.050	*C*_t_ = 6034.06e^−0.0203*t*^	0.9696	0.0203	34.13	−57.97
	0.10	*C*_t_ = 5991.31e^−0.0230*t*^	0.9759	0.0230	30.19	−52.48
	0.50	*C*_t_ = 5749.62e^−0.0437*t*^	0.9803	0.0437	15.85	−9.52
^2^ NO^3−^/mg ^L−1^	0.10	*C*_t_ = 6421.40e^−0.0603*t*^	0.9730	0.0603	11.49	24.81
	1.0	*C*_t_ = 6661.17e^−0.0508*t*^	0.9810	0.0508	13.66	5.03
	10	*C*_t_ = 7093.17e^−0.0783*t*^	0.9686	0.0783	8.86	61.96
	50	*C*_t_ = 7284.60e^−0.1124*t*^	0.9857	0.1124	6.17	132.60
^2^ NO^2−^/mg ^L−1^	0.010	*C*_t_ = 6270.93e^−0.0489*t*^	0.9911	0.0489	14.18	1.16
	0.10	*C*_t_ = 6840.22e^−0.0567*t*^	0.9817	0.0567	12.22	17.43
	1.0	*C*_t_ = 6602.46e^−0.0490*t*^	0.9864	0.0490	14.15	1.41
	10	*C*_t_ = 6681.25e^−0.0717*t*^	0.9933	0.0717	9.67	48.39
^2^ Tween80/CMC	1.0	*C*_t_ = 6199.92e^−0.0537*t*^	0.9865	0.0537	12.90	11.18
	2.0	*C*_t_ = 6420.75e^−0.0507*t*^	0.9932	0.0507	13.67	4.90
	5.0	*C*_t_ = 6549.50e^−0.0331*t*^	0.9804	0.0331	20.95	−31.52
	50	*C*_t_ = 6823.51e^−0.0190*t*^	0.9695	0.0190	36.44	−60.64
^2^ CTAB/CMC	0.10	*C*_t_ = 4786.30e^−0.0517*t*^	0.9597	0.0517	13.40	7.08
	0.50	*C*_t_ = 6448.72e^−2.3920*t*^	0.9843	2.3920	0.29	4850.43
	1.0	*C*_t_ = 5752.74e^−4.2599*t*^	0.9584	4.2599	0.16	8716.06
	10	*C*_t_ = 6234.73e^−3.5119*t*^	0.9867	3.5119	0.20	7168.03
^2^ SDBS/CMC	0.10	*C*_t_ = 6162.35e^−0.0392*t*^	0.9928	0.0392	17.66	−18.77
	0.50	*C*_t_ = 5639.97e^−0.0421*t*^	0.9697	0.0421	16.45	−12.79
	1.0	*C*_t_ = 6309.99e^−0.0403*t*^	0.9874	0.0403	17.21	−16.66
	10	*C*_t_ = 5923.69e^−0.0227*t*^	0.9798	0.0227	30.54	−53.02
^2^ Propyrisulfuron/mg L^−1^	1.0	*C*_t_ = 6234.50e^−0.0557*t*^	0.9837	0.0557	12.44	15.27
	10	*C*_t_ = 6336.75e^−0.0498*t*^	0.9654	0.0498	13.92	3.02
	100	*C*_t_ = 6450.54e^−0.0537*t*^	0.9838	0.0537	12.90	11.22
	500	*C*_t_ = 6461.64e^−0.0486*t*^	0.9736	0.0486	14.27	0.52
^2^ Humic acid/%	0.050	*C*_t_ = 6778.75e^−0.0665*t*^	0.9829	0.0665	10.43	37.56
	0.10	*C*_t_ = 6821.76e^−0.0705*t*^	0.9816	0.0705	9.83	45.92
	0.25	*C*_t_ = 6502.14e^−0.0582*t*^	0.9934	0.0582	11.90	20.53
	0.50	*C*_t_ = 5976.80e^−0.0349*t*^	0.9879	0.0349	19.88	−27.84
^2^ Biochar/%	0.050	*C*_t_ = 6350.65e^−0.0648*t*^	0.9719	0.0648	10.69	34.15
	0.10	*C*_t_ = 5646.26e^−0.0562*t*^	0.9759	0.0562	12.34	16.20
	0.25	*C*_t_ = 5040.18e^−0.0561*t*^	0.9565	0.0561	12.36	16.08
	0.50	*C*_t_ = 5147.28e^−0.0761*t*^	0.9591	0.0761	9.11	57.41

Where the influencing factors marked as “1” or “2” referred to the blank control group 1 or 2 as the references, respectively. PA represented polyamide, PHB represented poly-*β*-hydroxybutyrate, PS represented polyvinyl benzene, PBS represented polybutanediol succinate, PBAT represented butylene adipate and butylene terephthalate copolymer, LDPE represented low density polyethylene, PHA represented polyhydroxyalkanoates, PP represented polypropylene, PLA represented polylactice acid, PMMA represented polymethyl methacrylate, PVC represented polyvinyl chloride, PE represented polyethylene, DFMs represented disposable face masks, CMPF represented calcium magnesium phosphate fertilizer, OICF represented organic and inorganic compound fertilizer, CTAB represented cetyltrimethyl ammonium bromide, SDBS represented sodium dodecylbenzene sulfonate and *PR/IR* represented promoting or inhibiting ratio.

## Data Availability

Not applicable.

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
