# Peer review of "Degradation of a New Herbicide Florpyrauxifen-Benzyl in Water: Kinetics, Various Influencing Factors and Its Reaction Mechanisms"

_ijms, 2023, doi:10.3390/ijms241310521_

Round 1
Reviewer 1 Report
This manuscript reported the extensive study of effects of factors including temperature, pH, concertation, and environment factors in herbicide florpyrauxifen-benzyl degradation. The degradation products were also studied and mechanisms behind the degradation were explained. In general, the novelty of the idea is acceptable, the massive work is appreciated, experiments were designed well, results were discussed in a scientific manner, and the manuscript was drafted within a logistic structure. I suggest a minor revision with concerns listed below being addressed.
Comments:
1. Line number should be added into the manuscript for the convenience of editors and reviewers.
2. Full name should be present where acronyms are introduced for the first time in the manuscript. For example, MPs and DFMs.
3. Title of Section 2.5 should be ‘Data analysis’.
4. Number of formula 1- 9 should be aligned.
5. In Table 1, not all of the coefficient of determination (R2) value are close to 1, and thus the sound of ‘first-order kinetic’ may be weakened. Did the author try other equations or modify formula (1) in order to get a better correlation results?
6. Figure 3, last row of figures needs to be aligned well with others.
7. In reference section, line space should be narrowed down (and font size may also need to be adjusted) according to the template.
English is acceptable.
Reviewer 2 Report
The manuscript deals with the “degradation of a new herbicide florpyrauxifen-benzyl in water: kinetics, various influencing factors and its mechanisms”. It is very well written and the potential degradation of the active compound has been investigated through a very extensive prism.
Introduction: The first paragraph presents the serious problem of the contamination of the ecosystem with pesticide residues and their negative effects. Additional references should be added.
2.2. Preparation of buffer solutions: Please rephrase the whole paragraph and set the text in a correct grammatical order.
2.3. Hydrolysis test: Please delete “the” before quantities. Number (2) is also missing.
2.4.1. Preparation of standard and matrix-match standard working solutions: Please replace “matrix-match” with “matrix-matched” and describe the origin of each matrix used.
2.5.Dataanalysis: Please separate words.
Extensive editing of English language required
Reviewer 3 Report
Dear Sir,
The manuscript is interesting and the study well-conducted. However, there are some aspects that need clarifications and complements of information before the paper could be considered for acceptance.
The authors mentioned that “transformation and toxicology in the environment have seldom been reported”. Please make a comparison with Miller MR, Norsworthy JK (2018) Assessment of Florpyrauxifen-benzyl Potential to Carryover to Subsequent Crops. Weed Technol doi: 10.1017/wet.2018.33, who proposed a degradation pathway based on the degradation products collected in a real-time application of the herbicide.
For the separation of the acetonitrile-water solution (both solvents being miscible) by using NaCl and MgSO4 cite the study by Li et al., Salt-Induced Liquid–Liquid Phase Separation: Combined Experimental and Theoretical Investigation of Water–Acetonitrile–Salt Mixtures, J. Am. Chem. Soc. 2021, 143, 2, 773–784. Were the quantities of the two salts the optimal for complete extraction in the acetonitrile layer of all organic compounds?
In paragraph 3.2. mention also microbial activity, which is usually the primary degradation mechanism for pesticides.
The degradation pathways and resulting compounds are (some of them) quite puzzling, from an organic chemistry point of view.
For florpyrauxifen-benzyl:
Pathway 1 is correct, hydrolysis of the ester moiety is most likely to occur.
Pathway 2 is not possible – there is no way that the carbonyl of an ester group can be reduced in such manner. More probably the demethylation at the anisole moiety on the left aromatic ring is the process that occurs.
Pathway 3 is equally impossible. The loss of 45 could imply both demethylation and one hydrodechlorination (replacement of Cl by H) on the vicinal chlorine, in the left-side aromatic ring. Hydrodechlorination could occur on the influence of a sufficiently strong metallic cation, water playing the role of a proton donor.
Pathway 4 cannot occur for the same reasons. The compound that could be formed is more probably the result of a simultaneous demethylation, hydrodechlorination and hydrodefluorination on the same aromatic ring – the left one.
Pathway 5 could be possible on paper, although hydrodefluorination is rather difficult (more difficult than hydrodechlorination).
Pathway 6 could be a double hydrodechlorination on both aromatic rings.
For florpyrauxifen:
Pathway 1 is quite improbable, the reduction of the -COOH to the aldehyde in an aqueous media is not likely to occur. A demethylation of CH3-O-Ar is more probable.
Pathway 2 – the loss of the carboxyl group is more likely a decarboxylation with loss of CO2.
Pathways 3 and 4 are impossible.
In Pathway 3 a more probable loss would be demethylation associated with hydrodefluorination, In Pathway 4, the rupture of the biphenyl bond is simply impossible – the environmental fate of such compounds usually imply the advanced oxidation of one of the two aromatic rings (see for example R.E. Parales, K.-S. Ju, chapter 6.12 - Rieske-Type Dioxygenases: Key Enzymes in the Degradation of Aromatic Hydrocarbons, in Comprehensive Biotechnology (2nd Edition), Murray Moo-Young Ed., Academic Press, 2011, 115-134)
Pathway 5 could be indeed a hydrodechlorination, but I would have chosen the other chlorine, since the next step (Pathway 6) would be the decarboxylation (via elimination of CO2, not formic acid) of the vicinal -COOH moiety.
Each of these degradation steps are assumptions made by the authors. It would have been interesting if such processes could have been confirmed by similar process occuring on compounds that bear same functional groups, through literature references. Otherwise, the authors should clearly mention that they are only proposals for eventual degradation products.
To this date, the only confirmed degradation products of florpyrauxifen-benzyl that were indeed found are the 3 compounds that are present in the above-mentioned paper by Miller and Norsworthy.
For the reasons mentioned above I do not think that the manuscript is yet ready for acceptance, and needs a major revision in aspect to the degradation mechanisms discussion.
Some minor spelling and phrasing errors are present in the text. A careful re-reading for correcting these errors is requested.
Round 2
Reviewer 3 Report
Dear Sir,
The authors have answered my queries, with one exception. Please add the paper by Miller et al., since it is the only one that treats the problem of florpyrauxifen-benzyl degradation pathway, along the same explanation that they gave me in their answer. Include this reference and explanation in the 3.3. paragraph.
After that, the manuscript can be considered for acceptance.
The Language is of good quality and the manuscript readable.
